# Self-Organizing Maps for Clustering Hyperspectral Images On-Board a CubeSat

Aksel S. Danielsen, Tor Arne Johansen and Joseph L. Garrett *

Department of Engineering Cybernetics, Norwegian University of Science and Technology,
7491 Trondheim, Norway; akselus55@gmail.com (A.S.D.); tor.arne.johansen@ntnu.no (T.A.J.)
*   Correspondence: joseph.garrett@ntnu.no

**Abstract:** Hyperspectral remote sensing reveals detailed information about the optical response of a scene. Self-Organizing Maps (SOMs) can partition a hyperspectral dataset into clusters, both to enable more analysis on-board the imaging platform and to reduce downlink time. Here, the expected on-board performance of the SOM algorithm is calculated within two different satellite operational procedures: one in which the SOM is trained prior to imaging, and another in which the training is part of the operations. The two procedures are found to have advantages that are suitable to quite different situations. The computational requirements for SOMs of different sizes are benchmarked on the target hardware for the HYPSO-1 mission, and dimensionality reduction (DR) is tested as a way of reducing the SOM network size. We find that SOMs can run on the target on-board processing hardware, can be trained reasonably well using less than 0.1% of the total pixels in a scene, are accelerated by DR, and can achieve a relative quantization error of about 1% on scenes acquired by a previous hyperspectral imaging satellite, HICO. Moreover, if class labels are assigned to the nodes of the SOM, these networks can classify with a comparable accuracy to support vector machines, a common benchmark, on a few simple scenes.

**Keywords:** hyperspectral; ocean color; clouds; algae; imaging spectroscopy; Cubesat; AI4EO

## 1. Introduction

Hyperspectral imaging, also called imaging spectroscopy, is quickly becoming a powerful remote sensing tool. Several hyperspectral satellite payloads have already been launched, such as Hyperion, the Hyperspectral Imager of the Coastal Ocean (HICO), the Hyperspectral Imaging Suite (HISUI), and PRISMA, while many more are planned [1–6]. By resolving a continuum of spectral bands, often more than 100, hyperspectral data can be used to resolve spectral features more clearly than multispectral images [7].

Image processing in space is subject to computational constraints that are less applicable on earth, particularly data transmission constraints, constraints on computational performance, and constraints on storage. The Cubesat standard has enabled a proliferation of small satellites in the past two decades, and these satellites, owing to their relatively small power budgets, as well as their size and cost limitations, suffer notably from these constraints [8]. As hyperspectral imagers (HSIs) are passive sensors—unlike lidar or radar—their power demands are appropriate for a cubesat. However, because any image processing performed on-board competes with the radio communication system and the imager itself for resources, the computational budgets can be limited. The first HSI cubesats, Aalto-1 and Hyperscout, were launched in 2017 and 2018, respectively, and the latter incorporates on-board processing, validating the concept [9,10].

The HYPSO-1 (HYPer-spectral Smallsat for ocean Observation) CubeSat is being developed by NTNU Small Satellite Lab and will launch in December 2021 [11]. The primary goal of the mission is to observe ocean color along the Norwegian coast. It will target harmful algal blooms, among other phenomena. HYPSO-1 will be the apex of a

network of autonomous agents monitoring the coastline and will provide the other agents with a synoptic view of the ocean to aid with their own mission-planning [12].

The computational limitations of the platform present difficulties for the mission. The radio bandwidth is low relative to the size of the captured HSI data, which means that if the whole data cube is downlinked, the autonomous agents will be unable to respond to dynamic phenomena in anything resembling real-time. Lossless compression can reduce the size of the data by more than a factor of 2 [13], but this reduction is not sufficient to achieve real-time behavior. Moreover, compressed data are not suitable for further analysis on board. Clustering can achieve a significant reduction in the size of the data, while retaining it in a suitable form for further processing in tasks such as target detection or classification.

### 1.1. Related Works

Clustering, in the context of hyperspectral imaging, refers to grouping similar pixels together, without regard for what the pixels represent in the scene. This is unlike classification, the more common activity of grouping pixels together with the surface constituent (e.g., water or land), which led to a particular spectrum, because it does not require information about the ground truth, which is often impossible to acquire for hyperspectral images. However, both aim to assign each input vector in a dataset to one of a finite number of discrete categories [14,15]. Clustering can even be used as pre-processing for classification, so that the latter algorithm classifies clusters rather than the pixels themselves. This is often how clustering algorithms are evaluated and compared.

Centroid-based algorithms represent each cluster by its center. *k*-means [16], one of the most popular centroid based clustering methods, is known for its simplicity and computational efficiency. It has been applied to hyperspectral data with considerable success in both reducing the time required for classification and increasing its overall accuracy, both as a pre-processing method for classification by SVMs or combined with spatial information [17,18].

The self-organizing map (SOM) [19] is closely related to to *k*-means, with a lattice placed between the nodes in order to impose a distance metric, although it also has connections to the gaussian mixture model discussed below [20]. SOMs were originally developed to model biological neural networks [21], but have been used in hundreds of different applications [19,22], including compression [23] and regression [24].

Early experiments that applied SOMs to hyperspectral data using small network sizes showed they were potentially valuable for classification and target detection [25,26]. Starting in 2018, Riese et al demonstrated that SOMs can be used for regression to determine parameters which vary nonlinearly, with performance comparable to more common machine learning methods [27,28]. Later, they showed that that SOMs enable the classification of hyperspectral images with very few labelled pixels and presented their own framework, which accommodates supervised, semi-supervised, and unsupervised learning with SOMs [29]. At the same time, experiments showed that pre-processing with sufficiently large SOMs facilitates levels of classification accuracy exceeding Support Vector Machines (SVMs) [30]. In addition, the ease of visualizing SOMs has encouraged their use in a number of other tasks related to the analysis of satellite imagery [31–34].

Spectral Clustering (SC) [35] non-linearly transforms data into a lower-dimensional space before clustering in the lower dimension using another procedure, such as *k*-means. As generic spectral clustering has computational requirements that are too demanding for hyperspectral images, several customized algorithms have been designed, notably Fast Spectral Clustering with Anchor Graph (FSCAG) and fast spectral clustering (FSC) [36,37]. These algorithms utilize anchor points, rather than every pixel, when constructing the affinity matrix, and the latter uses non-negative matrix factorization to simplify the calculation further [38].

Model-based clustering algorithms partition data by creating a probabilistic model to describe how the spectra of pixels are generated. Each pixel is considered to be a random

variable drawn from a distribution that depends on latent, unobserved variables. The clustering problem is then transformed into an inference problem of estimating the latent labels from the observed data. For hyperspectral images, this is often based on the gaussian mixture model (GMM) solved by the Expectation Maximization (EM) algorithm [39]. The GMM models the image as several multivariate Gaussian distributions, from which the pixels are drawn. It has been used to cluster hyperspectral images and as preprocessing for classification [40,41]. An extension of this algorithm for hyperspectral data, called a Label-dependent Spectral Mixture Model with Markov Random Field (LSMM-MRF), is proposed in [42].

The on-board processing of hyperspectral images for classification and target detection often requires algorithms to be adapted, both to achieve a real-time performance [43,44] and to accommodate the resource-constrained environments that are available on smaller platforms [45,46]. For example, Du and Nekovei proposed using only a subset of the pixels at several steps in an algorithm [43], while Thompson et al. demonstrated a way of accommodating sensor abberations without any algorithm to correct them by applying a target detection algorithm columnwise [47]. Other research, such as a recent set of experiments by Alcolea et al., has focused on determining which pre-existing algorithms are best suited to on-board operation [48].

### 1.2. Proposed Contributions

In this article, we present a strategy for using Self-Organizing Maps (SOM) to cluster hyperspectral images on a satellite. The SOM machine-learning technique is chosen because it has relatively low memory and processor requirements and clearly separates training and execution so that the algorithm can be split between the ground segment and the spacecraft (see Appendix A).

We develop and test SOMs on the HYPSO-1 hardware, and use those performance tests to derive size requirements based on execution time. Two operational procedures for the use of SOMs are evaluated. In one of the operational procedures, an SOM trained on one scene is applied to another. In the other, the SOM is trained on a very small subset of the pixels. Although motivated by the experiments of Riese et al. [29], the second procedure differs from these experiments because the missing pixels are not available at all, rather than simply being unlabelled. It is found that SOMs can be executed on the satellite within the computational time restraints, and the resulting clustering retains enough information for further image-processing, such as classification. Moreover, SOMs can be trained from a small portion of the total pixels in an image without a notable increase in the quantization error.

### 2. Materials and Methods

#### 2.1. Evaluation Data

Different datasets are used to evaluate the performance of the SOMs with respect to unsupervised and supervised learning. Unsupervised learning is used to evaluate how well a SOM can approximate a dataset. Supervised learning, as a classification method, is used to evaluate how much information is retained in the SOM clustering for subsequent analysis.

The unlabelled data come from the Hyperspectral Imager of the Coastal Ocean (HICO) instrument, which was hosted on the International Space Station [1]. The data are used as L1 Top of atmosphere (TOA) radiance, so that the clustering does not depend on a particular atmospheric correction. Three different locations are chosen to evaluate the algorithm (Table 1 and Figure 1). First, the North Sea is chosen because HICO observed that region many times, and it is relatively close to the HYPSO-1 target region of the Norwegian coast. Second, the Pacific ocean along the coast of Chile is also chosen because it is also subject to harmful algal blooms. Third, Lake Erie, along the Canada-USA border, is chosen because it suffers frequent harmful algal blooms [49] and, in contrast to the first two locations, is freshwater.

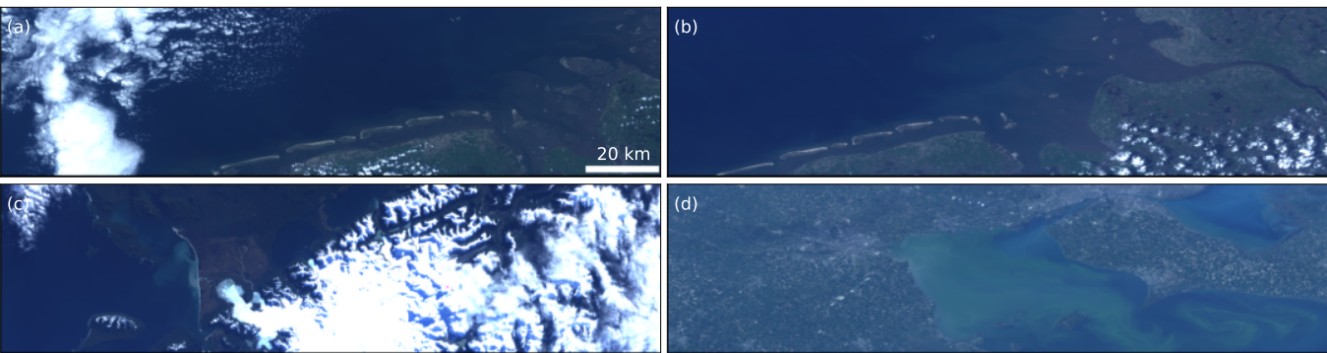

**Figure 1.** The four HICO scenes used in the analysis: first North Sea scene (**a**), second North Sea scene (**b**), Laguna San Rafael—Chile (**c**), Lake Erie—USA/Canada (**d**).

**Table 1.** Unlabelled Data, from HICO.

| Location Name | Latitude | Longitude | Date |
|---|---|---|---|
| North Sea *—Germany | 53.9°N | 7.3°E | 2 September 2014 |
| North Sea—Germany | 53.9°N | 8.1°E | 1 May 2013 |
| Laguna San Rafael—Chile | 46.7°S | 74.7°W | 13 June 2012 |
| Lake Erie—USA/Canada | 41.8°N | 83.1°W | 3 September 2011 |

* used for initial analysis.

Although these data are initially unlabelled, during this study, one image from the North Sea is labelled as cloud, land, and water. This labelling is imprecise in order to approximate the accuracy of the labelling that will be possible during HYPSO satellite operations, when time is constrained. In other words, amateur operators aided by a map should be able to distinguish the three categories, except perhaps in a few ambiguous cases (e.g., foggy swamp). Each image contains $\approx 10^6$ pixels and 86 bands (410–897 nm range, 5.7 nm bandwidth).

The labelled data originate from four hyperspectral datasets commonly used to evaluate classification algorithms: Pavia University, Indian Pines, Samson, and Jasper Ridge. The Pavia University scene was acquired by the ROSIS sensor [50] and the Indian Pines scene was acquired by the AVIRIS sensor [51], with preprocessing described in [52]. In each scene, a subset of the pixels is classified into 9 or 16 classes, respectively. The latter two images, described in [53], each pixel is partitioned into different endmembers. The class of each pixel is then defined as the endmember with the largest weight.

### 2.2. The HYPSO-1 Payload

The sensor used in the hyperspectral imager on HYPSO-1 is the Sony IMX249 from IDS Imaging Development Systems GmbH [54]. The standard cube size on HYPSO-1 is $(h, w, d) = (956, 684, 120)$, where $h$ is the number of captured frames, $w$ is the number of rows in each frame while $d$ is the number of binned spectral bands in the cube. Therefore, there are, in total, $n = h \times w \approx 6.5 \times 10^5$ pixels in a cube.

The On-Board Processing Unit

The HYPSO payload is controlled by a System on Chip (SoC), the Xilinx Zynq-7030. It consists of the Processing System (PS) and a Kinex-7 based Programmable Logic (PL) module. The main technical specifications for the PS are listed in Table 2. The PS will handle the main payload software tasks such as image capturing, image processing, communications, and file transfers. To accelerate the heavy workloads, several algorithms will be synthesised and programmed to the PL.

**Table 2.** On-board Processing Unit technical specifications

| Component | Specification |
|---|---|
| **Zynq-7030** | |
| Processor | 32-bit Dual Core ARM Cortex-A9 @ 667 MHz |
| Architecture Extensions | NEON 128b SIMD coprocessor |
| L1 Cache | 32 KB per processor |
| L2 Cache | 512 KB shared |
| On-Chip Memory | 256 KB |
| **Other system memories** | |
| SDRAM | 1 GB DDR3 |
| QSPI Flash | 128 Mb |
| eMMC | 8 GB |
| SD card | 16 GB |

The SoC is located on the PicoZed System on Module (SoM). It features 1 GiB of DDR3 SDRAM, which is the main working memory for the pipeline. A custom-designed breakout board is connected to the PicoZed for IO management. In addition, it contains temperature sensors and a heat sink.

### 2.3. Image Processing Pipeline

The on-board processing framework is a software framework that allows for the processing of hyperspectral cubes in a pipeline. Each module applies an algorithm to the cube in series. Some modules, such as the smile and keystone correction [55] and compression [13], modify the cube, as it propagates through the pipeline. Other modules, such as target detection [56] and classification, analyze the cube.

#### 2.3.1. Dimensionality Reduction

Principal Component Analysis (PCA) is used as the DR because it is a common DR technique, its only hyperparameter is number of dimensions, and the components it produces are orthonormal, which enable simple inversion [57,58]. The determination of the PCA components occurs on the ground, so that only the projection of the image onto the components to calculate the scores (spatial distribution of weights) must be performed on the satellite. In this paper, the components are initially computed using the scikit-learn library [59]. For the SVM classification, the data are normalized by their variance before performing the PCA transform.

#### 2.3.2. Classification

Support vector machines with a radial basis functon (RBF) kernel are used as the benchmark to compare to SOM-based classification because they are the state-of-the-art technique, which, like SOMs, does not incorporate spatial information. They have a small memory footprint suitable for embedded systems, although they are somewhat computationally intensive [48,60]. Scikit-learn is used for the accuracy analysis [59] while speed benchmarking is performed on the target hardware with custom code written in *C*.

### 2.4. Operations

Contextualizing an algorithm within an operational procedure can reveal hidden constraints. Therefore, two operational plans are presented, in which the SOM is used for clustering (Figure 2). In both, the main purpose of clustering is compression. A secondary purpose is to pre-process the image for classification into land, cloud, and water, the output of which can either be sent to autonomous agents for navigational purposes or used in other algorithms, such as georeferencing.

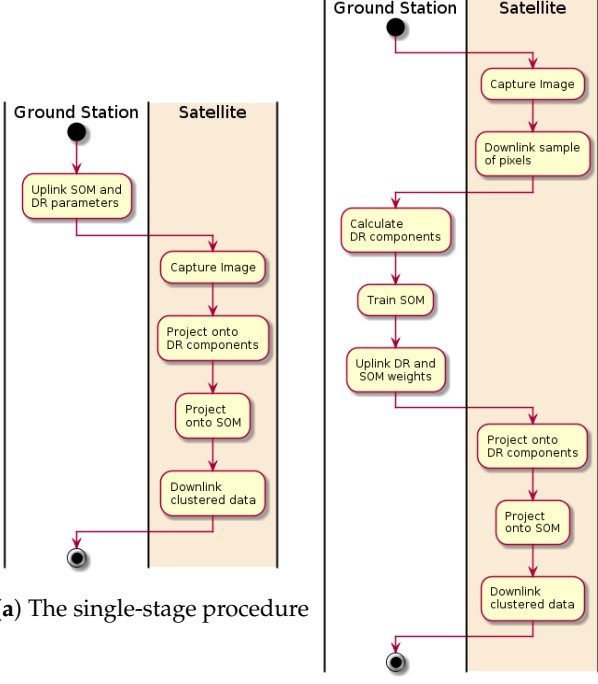

(**a**) The single-stage procedure

(**b**) The two-stage procedure

**Figure 2.** The two proposed operational procedures, which use Self-Organizing Maps to compress HSI data.

The first operational procedure uses a pre-calculated SOM to cluster an image (Table 3). The difficulty of this plan is that the captured image might not be sufficiently similar to the image from which the SOM is calculated. The second procedure calculates the SOM from the captured image (Table 4), by downlinking a random selection of pixels. At the ground station, the PCA components are calculated and the SOM is trained from the selection of points. The relevant DR components and the SOM are then uplinked back to the satellite, before the image is clustered and downlinked.

There are two main difficulties with the latter plan: (1) it is operationally more complex and (2) an SOM must be trained with only a small selection of pixels. Difficulty (2) is investigated in this article, while (1) is expected to be overcome with operator training and planning.

**Table 3.** The single-stage procedure for data clustering.

| Action | Size (kB) | Duration (s) |
|---|---|---|
| 1. Uplink PCA loadings | 1.2 | 0.24 |
| 2. Uplink SOM weights | 11.0 | 3.00 |
| 3. Capture image | 160,000.0 | 53.00 |
| 4. Projection onto DR components | 6500.0 | 5.00 |
| 5. Project onto SOM | - | 54.00 |
| 6. Downlink clustered image | 1100.0 | 11.00 |

**Table 4.** The two-stage procedure for data clustering.

| Action | Size (kB) | Duration (s) |
|---|---|---|
| **First stage** | | |
| 1. Capture image | 160,000.0 | 53.00 |
| 2. Downlink random selection of pixels | 250.0 | 2.00 |
| 3. Perform PCA at ground station | - | - |
| 4. Train SOM at ground station | - | - |
| **Second stage** | | |
| 5. Uplink PCA loadings | 1.2 | 0.24 |
| 6. Uplink SOM weights | 11.0 | 3.00 |
| 7. Projection onto DR components | 6500.0 | 5.00 |
| 8. Project onto SOM | - | 54.00 |
| 9. Downlink clustered image | 1100.0 | 11.00 |

The time taken to project onto the SOM is calculated from the timing benchmarks determined below. The times are calculated assuming a $32 \times 32 \times 5$ SOM, a selection of 1024 pixels, an uplink bandwidth of 0.4 Mbps, and a downlink bandwidth of 1.0 Mbps. The SOM-clustered data are assumed to use an unsigned 2-byte integer, although the cluster labels can be expressed with only 10 bits, because the former is a more common data format. In both tables, size and memory are both rounded up to two significant figures.

*2.5. Self-Organizing Maps*

A SOM maps input data to a low-dimensional space, typically consisting of far fewer datapoints than the original data. This is done by training a lattice of nodes that preserves the distribution of the input data while also representing the full dataset. While the lattice can have any structure, the most common is a finite two dimensional rectangular grid.

Each node of the SOM is represented by vector $\mathbf{z}$ with the same dimension as the input data points. The operation of the SOM consists of two parts: training and labelling. Here, we discuss the training and labelling separately, although we note that they can be combined so that the SOM is trained on-line. Both parts involve finding the best matching unit (BMU) for an input vector, or the node which is the closest to the input vector under a given distance metric, typically Euclidean. In other words,

$$\mathrm{BMU}(\mathbf{x}_i) = \underset{\mathbf{z} \in \mathbf{Z}}{\mathrm{argmin}}\, k(\mathbf{x}_i, \mathbf{z}), \tag{1}$$

where $\mathbf{X} = \begin{bmatrix} \mathbf{x}_1 & \mathbf{x}_2 & \ldots & \mathbf{x}_n \end{bmatrix}$, $\mathbf{x}_i \in \mathcal{R}^d$ are the input data, $\mathbf{Z} = \begin{bmatrix} \mathbf{z}_{1,1} & \ldots & \mathbf{z}_{m,m} \end{bmatrix}$, $\mathbf{z}_i \in \mathcal{R}^d$ are the $m \times m$ grid of SOM nodes, and $k$ is the distance metric. In the training phase, a neighborhood function of the BMU is calculated and the nearby nodes are adjusted towards the input vector, while in the labelling phase, the input vector is labelled with the lattice coordinates of the BMU (illustrated in Figure 3). Once the map is trained, any two points that are close (with respect to $k$) in the input data space are mapped to spatially close nodes in the SOM grid. Therefore, the grid preserves the topology of the input space.

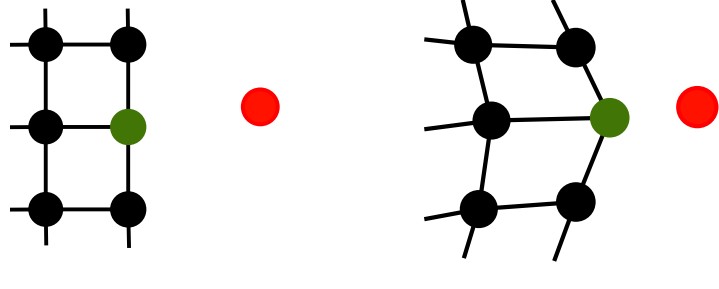

(**a**) Before training (**b**) After training

**Figure 3.** Training the SOM on a new input sample (**red**). The best-matching unit (**green**) is affected the most, while the neighboring nodes are less influenced. The distance between the nodes is proportional to the similarity in the data space.

The training proceeds similarly to gradient descent, although, because an objective function for the original SOM has not been found, it is not technically gradient descent [61]. Let $\mathbf{u}_i^* = \mathrm{BMU}(\mathbf{x}_i)$ denote the index on the SOM of the BMU for input vector $\mathbf{x}_i$ from Equation (1). The update step $s$ for each SOM node $\mathbf{z}_j$ is given by

$$\mathbf{z}_j^{s+1} = \mathbf{z}_j^s + \alpha\beta(t, \mathbf{u}_j, \mathbf{u}_i^*)k(\mathbf{x}_i, \mathbf{z}_j^s),\qquad(2)$$

where $\alpha > 0$ is the learning rate for iteration, $\beta(t, \mathbf{u}_j, \mathbf{u}_i^*) > 0$ is the neighborhood function, $t$ denotes the epoch of training, and $s$ denotes the iteration within the epoch.

The learning rate $\alpha$ determines how fast the map adapts to the input data. The neighborhood function $\beta(t, \mathbf{u}_j, \mathbf{u}_i^*)$ preserves the topology by training nodes that are spatially far from $\mathbf{u}_i^*$ with a lower magnitude. It is characterized by an update radius $\sigma(t)$, which decreases as the training progresses in order to initially learn the rough distribution topology before finetuning local areas of distribution.

### 2.5.1. Structure

The discussed SOMs are all on a rectangular lattice (e.g., a grid), rather than a hexagonal lattice, or some other lattice structure. In addition, all the grids are square in the sense that they have the same number of nodes on each side. These structural parameters are chosen so that the focus of the article is how well an SOM can perform within the two operational procedures, rather than how to optimize an SOM architecture, which has already been investigated in numerous other studies [19,22].

### 2.5.2. Initialization

The SOMs are initialized by computing the first two principal components of the data. The upper-left and lower-right corners are initialized, respectively, with the spectra of the pixels that have the largest and smallest scores on the first principal component. The upper-right and lower-left corners are initialized with the spectra of the remaining pixels that have the largest and smallest scores on the second principal component. This initialization procedure is similar to the one proposed in [19], but uses the spectra of pixels in the data rather than directly using the principal components.

### 2.5.3. Training

All the training is performed using the gaussian neighborhood function. A learning rate of 0.1 is used throughout, while the update radius varies and is specified in each individual experiment. The order in which the pixels are input into the SOM is randomized.

### 2.5.4. Metrics

The main metric which is used to evaluate the performance of the SOMs is the relative quantization error for a pixel:

$$q_i = \sqrt{\frac{\left|\mathbf{x}_i - \mathbf{B}(\mathbf{x}_i)\right|^2}{|\mathbf{x}_i|^2}}, \tag{3}$$

where $\mathbf{B}()$ is the function that maps each spectrum to its BMU on the SOM, and $q_i$ is the relative quantization error for a single pixel. The denominator in (3) is included to allow the metric to be compared between scenes. Without this, using the standard quantization error from [19], the mean quantization error is strongly influenced by the number of pixels in a scene which are occupied by clouds because they are so much brighter than the other types of pixels. When using DR, it is assumed that $\mathbf{x}_i$ and $\mathbf{B}$ are in the original spectral representation, which is positive definite, in Equation (3). In the section which evaluates the effect of noise on the SOM, the relative quantization error is called the relative mean square root error (and $\mathbf{B}$ is replaced with the noisy data) because the name *quantization* is not relevant in that context.

In the section on classification, the overall accuracy is used:

$$\text{OA} = \frac{N_\text{C}}{N_\text{L}}, \tag{4}$$

where $N_\text{L}$ is the total number of labelled samples and $N_\text{C}$ is the number correctly labelled by the classification algorithm.

### 2.5.5. Node Labels

SOMs on their own are unsupervised networks and are not directly used for classification. Instead, the nodes are labelled to perform classification. The procedure used here for labelling the nodes is inspired by [29]; the stochastic parts of the training have been removed, although a probabilistic interpretation remains.

First, the SOM nodes are trained using all of the available data, both labelled and un-labelled, as above. Second, while all the spectra of the nodes are held constant, each labelled pixel is matched to its BMU. The number of times that any pixel from a class is assigned to a node is counted. A probability map is then estimated for each class by multiplying the counts at each node by the neighborhood function, summing them together, and normalizing them to 1. This procedure expresses the intuition that proximity on the SOM corresponds to spectral proximity. Finally, the nodes are categorized according to which class has the greatest probability at a given node, essentially a maximum likelihood interpretation. The update radius used in the calculation of the probabilities is chosen to be $r \approx \sqrt{z^2/(M\pi)}$, where $M$ is the number of labelled samples, so that the area covered by the neighborhood functions approximates the area of the SOM.

## 3. Results

### 3.1. Computational Benchmarks on Target Hardware

The computational time of each algorithm scales with its parameters, e.g., the SVM scales with the number of support vectors. To find parameters under which the models are computationally tractable within the constraints, they are tested on the target hardware (Figure 4). Then, based on the measurements, runtime models are interpolated. They are interpolated with only the highest-order term in the asymptotic runtime to provide a simple equation. The runtime analysis is based on the standard cube size of 956 frames, with 684 pixels, and 120 bands each). The ideal and maximum computational times are 190 and 380 s, respectively, to meet the demands of HYPSO's power budget.

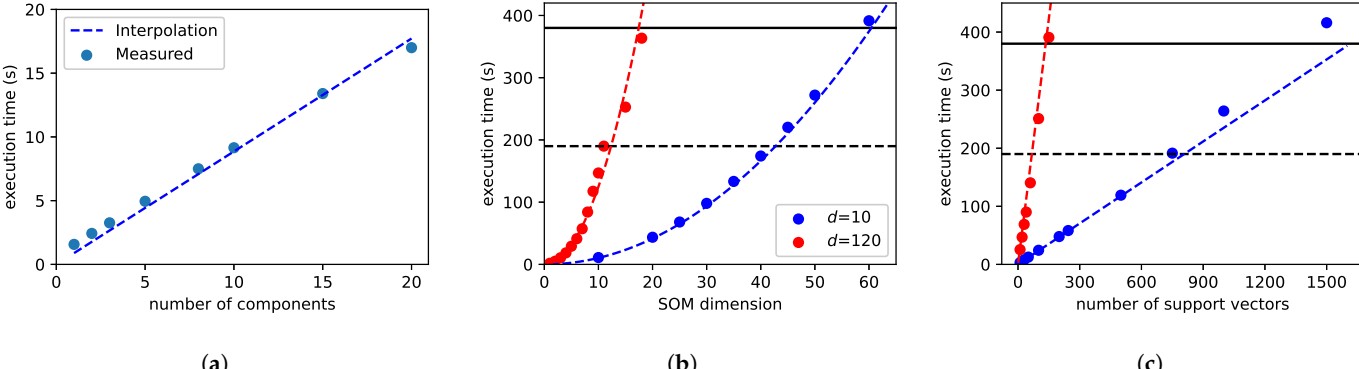

**Figure 4.** Execution time for PCA, SOM and SVM for a standard image with $(h, w) = (956, 684)$. The ideal requirement and hard requirement on computational time are shown with dashed and solid lines respectively. (**a**) Execution time for PCA projection; (**b**) Execution time for the SOM method; (**c**) Execution time for SVM.

First, the PCA DR projection is tested. The computational time of projecting a cube is expected to scale linearly with the number of projected components $d$. Figure 4a shows measurements of the computational time versus the number of components, as well as the interpolation. The resulting expected computational time in seconds is $E_{PCA} = 0.8858d$, or about 1.35 µs per pixel per component.

The computational time of the SVM is dominated by the RBF kernel calculation, which is $O(dn_s)$ where $n_s$ is the number of support vectors. From the measurements and corresponding interpolated model, the expected computational time is $E_{SVM} = 0.0235dn_s$. Figure 4c shows the interpolated model for measurements with $d = 10$ and $d = 120$ bands. The measurements mainly support the linear model; however, there is some deviance for a large number of support vectors. For the ideal requirement of <190 s of computation and a cube with full dimension $d = 120$, the upper limit on the number of support vectors is about 70, which is typically far below the number of support vectors in a model. Reducing the dimension to $d = 10$ gives time for about 750 support vectors.

The SOM method has a quadratic runtime in the dimension $z$ and the driving quadratic term also scales with the number of bands $d$. The interpolated model, $E_{SOM} = 0.0104\,dz^2$, is as seen in Figure 4b. Without DR, $z$ must be 11 or less to fit within the target computational time. Reducing the dimensionality to $d = 5$ allows $z$ to be about 56.

### 3.2. Clustering

In this section, the SOMs' ability to be trained on unlabelled data is evaluated. As the data acquired in the HYPSO-1 mission are initially unlabelled, this is a critical ability for SOMs in both operational plans. First, the structure of an SOM trained on the first North Sea scene without DR is investigated. Second, several SOMs are trained and evaluated on the first scene with DR. Next, the performance of transfer learning is tested by comparing SOMs that are trained and applied to different datasets to SOMs that are trained and applied to a single dataset. Finally, an SOM's capacity to deal with noisy data is evaluated.

Before the process of optimizing a SOM architecture for the HYPSO-1 mission, the structure of a single SOM and its application to unlabelled data are explored in depth. The first SOM is trained using the full 86-band spectrum from the first North Sea HICO scene (Figure 1a). The SOM consists of $64 \times 64$ nodes and is trained with an update radius decreasing from 32 to 1 and a constant learning rate of 0.1, in 100 steps of $10^4$ pixels at a time in a randomized order.

The initial SOM, while too large for on-board processing, because it lacks DR, gives a benchmark for SOM performance, and is interpretable in the sense that each node of the SOM can be represented as a spectrum. In Figure 5a, the nodes of the trained SOM are colored according to the spectrum which each node represents (normalized to the maximum of each band). The nodes of the upper left corner are white, corresponding to

cloud pixels, while the nodes of the upper right corner appear red, corresponding to the large infrared response of land pixels. Along the bottom, the nodes transition from a lighter blue on the left to a darker blue on the right, which corresponds to different spectra that water appears to have in the scene.

The U-matrix (Figure 5b) shows the distance of nodes from their neighbors, so that folds in the SOM appear as dark lines. Two folds are apparent: a vertical one extends down from the top and separates the nodes associated with clouds on the left from nodes associated with nodes associated with land, while a horizontal one extends from the right side and separates the land-pixels from the water pixels. The number of pixels in the original image that are identified with a particular node are plotted (logarithmically) in Figure 5c. The counts are distributed fairly uniformly over the SOM, except along the folds that are visible in the U-matrix, which indicates that the SOM expresses the content of the scene. The mean and median relative quantization error for the scene are 0.87% and 0.63%, respectively, which further indicates that the SOM has captured the pixel distribution of the scene.

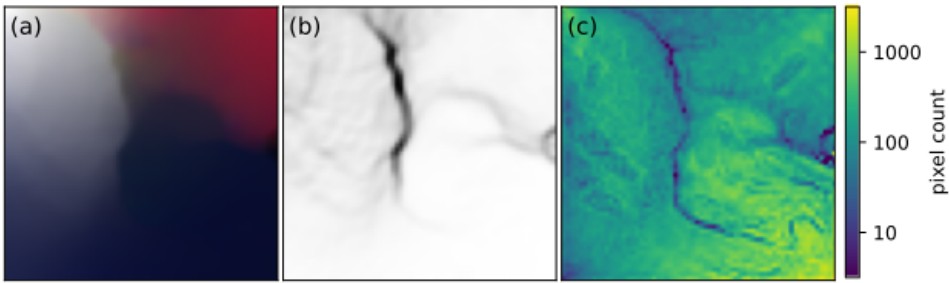

**Figure 5.** The SOM trained on the first North Sea scene. (**a**) The SOM in false color, represented by bands 816 nm (red), 684 nm (green) and 467 (blue). (**b**) The U-matrix shows the curvature of the surfaces at each node, where darker colors indicate more curvature (arbitrary units). (**c**) The number of pixels in the original image which are classified at each node.

The physical interpretation of the SOM is apparent from both the spectra of the nodes and the spatial distribution of pixel clusters in the scene. In Figure 6, the spectra for the nodes of the SOM are shown, with the spectra of the nodes in the four corners highlighted. The clustering of the first North Sea image is shown in Figure 7, with (a) showing the column number and (b) showing the row number.

The information depicted by the labels corresponds with what can be visually observed in the original color image (Figure 1). First, the regions that are bright in both (a) and (b) appear to be clouds, which is consistent with the observation that they are most similar to the black line in Figure 6, which itself has characteristics of the spectrum of a cloud (e.g., relatively bright and spectrally smooth). Likewise, the regions dark in (a) but bright in (b) appear to be land, consistent with the red-colored spectral curve. The regions that are dark in both seem to be water, which is consistent with the blue- and purple-colored spectra. Finally, Figure 7c, which depicts the per-pixel relative quantization, shows that the error is relatively larger on land than on either clouds or water. It is suspected that the SOM learns the spectra of the land pixels less well because they are more diverse.

Dimensionality reduction is investigated as a way of reducing the computational requirements of the SOMs. It is first tested by reducing the number of bands used in the training of the above SOM from 86 to 5. As the evaluation of a SOM is linear in the number of bands, this corresponds roughly to a speed improvement of about $17\times$. The relative quantization error only increases to a mean and median of 1.0% and 0.8%, or by about 20% relative to using all bands.

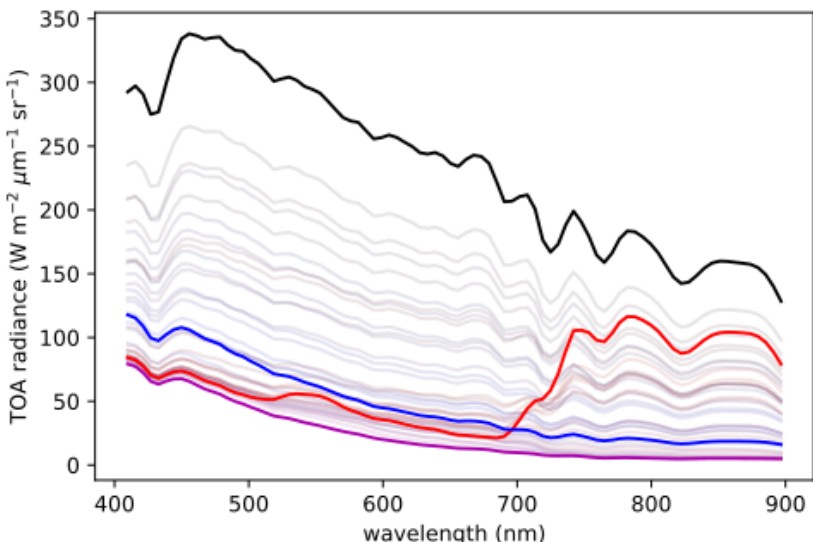

**Figure 6.** The spectra of every fourth node along each axis. The spectra in the corners are bold and colored black, red, purple, and blue (clockwise from the top left).

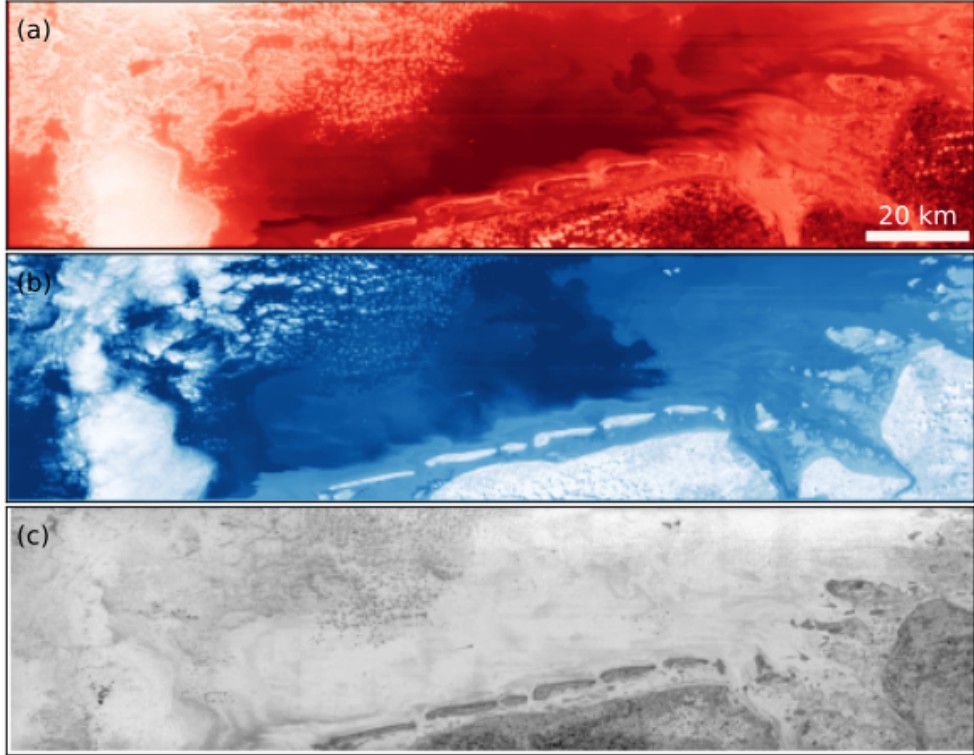

**Figure 7.** The first North Sea image classified according to the $64 \times 64 \times 86$ SOM. The (**a**) column and (**b**) row of the BMU index. (**c**) the relative quantization error.

The performance of DR is further evaluated by considering how SOMs of several different sizes with varying numbers of dimensions performed on the first North Sea scene (Figure 8). It is found that, in general, the quantization error decreases with both an increasing number of dimensions and SOM network size. However, for a given network size, there is a critical number of dimensions, above which there is no further reduction in the quantization error. For these reasons, the rest of the paper will work with five PCA dimensions, unless otherwise noted.

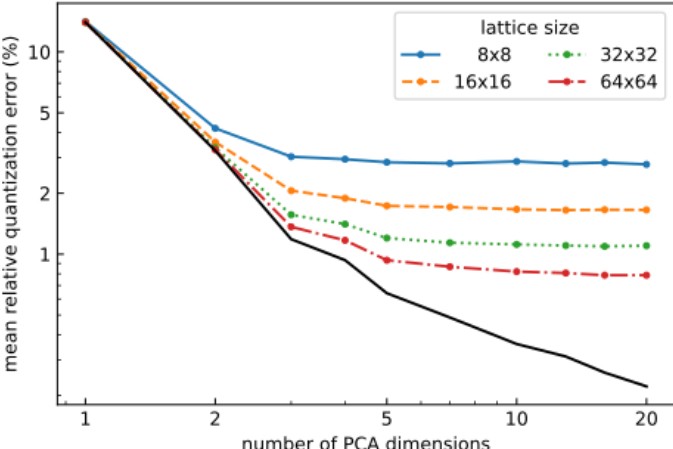

**Figure 8.** The mean quantization error for four different SOM sizes, for varying numbers of PCA-reduced dimensions. The black line depicts the error imposed by DR itself (prior to applying the SOM), and is thus a lower limit on the error.

The second operational procedure (Table 4) relies on training an SOM with a limited number of downlinked pixels. Therefore, the ability of a SOM to learn from a small amount of data is tested. A $32 \times 32 \times 5$ SOM is trained with an update radii of 8, 2.8 and 1 and a number of pixels ranging from $2^4$ to $2^{15}$. The training is performed as it would be during operations: both the determination of the PCA components and the training of the SOM are performed with a limited number of samples. Then, the SOM is evaluated on the full image. The procedure is repeated four times and the averages are presented in Figure 9.

For a given number of samples, the performance does tend to improve as the update radius is decreased, as is found above. For a given update radius, the performance improves with an increasing number of samples up to a certain value, $n^*$, above which it remains constant. Below $n^*$, the training data significantly underestimate the error, while above this, the training data provide a decent estimate of the error. These results validate the two-stage operational procedure.

The first operational procedure (Table 4) relies on transfer learning, particularly the application of an SOM trained on one image to a second image. This ability is tested by applying an SOM trained on the first North Sea image to the other images. It is found that the quantization error in the clustering of an SOM on a different training image results in a notable increase ($10 - 20\times$) relative to the image that it was originally trained on (Tables 5 and 6). The different scenes on which the SOM is applied lead to significantly different quantization errors. In addition, the SOM performed better on the freshwater scene (Erie) than it did on the same location on which it was trained (North Sea), which suggests that it is difficult to anticipate how well an SOM will perform on a particular scene prior to its application.

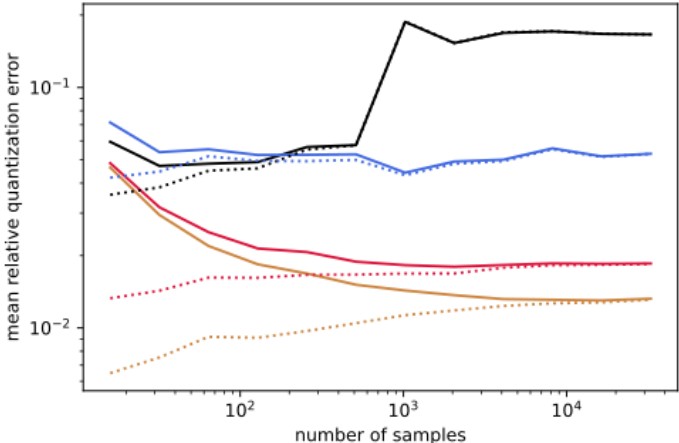

**Figure 9.** The mean quantization error as a function of sample size for PCA-initialized $32 \times 32$ SOM (**black**), or SOMs trained with an update radii of 8 (**blue**), 2.7 (**red**), and 1 (**orange**) as evaluated on either the reduced sample size (**dotted**) or full image (**solid**).

**Table 5.** Relative quantization error resulting from $64 \times 64 \times 86$ SOM classification trained by North Sea 1.

| Scene | Mean | Median |
|---|---|---|
| North Sea 1 | 0.0087 * | 0.0063 * |
| North Sea 2 | 0.0733 | 0.0683 |
| Laguna San Rafael | 0.1560 | 0.1372 |
| Lake Erie | 0.0614 | 0.0578 |

\* training scene.

**Table 6.** Relative quantization error resulting from $32 \times 32 \times 5$ SOM classification, either trained on North Sea 1, or from subsampling one scene with 4096 pixels.

| Scene | Transfer | | Subsampled | |
|---|---|---|---|---|
| | Mean | Median | Mean | Median |
| North Sea 1 | * | * | 0.0129 | 0.0096 |
| North Sea 2 | 0.0766 | 0.0780 | 0.0152 | 0.0119 |
| Chile | 0.2209 | 0.1775 | 0.0241 | 0.0186 |
| Erie | 0.0710 | 0.0678 | 0.0160 | 0.0132 |

\* used for training in transfer learning test.

The performances of the two operational procedures, in terms of the relative quantization error of the resulting datasets, are compared by applying each to the test datasets. A $32 \times 32 \times 5$ SOM is trained from a random sampling of 4096 pixels on each test image, and the trained SOM is tested on the same image (Table 5). Then, the SOM trained on the first North Sea scene is applied to the other scenes.

For all the scenes, the SOM trained on a sub-sampling of pixels resulted in smaller relative quantization errors 4–10× than when transfer learning is used. The SOM transferred to the Lake Erie scene shows that while the SOM captures the algal bloom, it does not resolve the gradations in the chlorophyll content, and has much more relative error over the whole scene compared to the first North Sea image, particularly over the water (Figure 10). On the other hand, the SOM trained on a sub-sampling of the Lake Erie pixels shows the algal bloom quite clearly. The relative quantization error over the bloom is mostly less than 1%. The Laguna San Rafael scene gives the largest quantization error for both procedures, perhaps because the scene contains diverse surfaces, including fresh water, ocean water, mountains, glaciers, swamps, and clouds (Figure 11). While the second North Sea scene gives a higher quantization error than the Lake Erie scene under transfer learning, it gives a lower error than the latter when the sub-sampled SOM is applied.

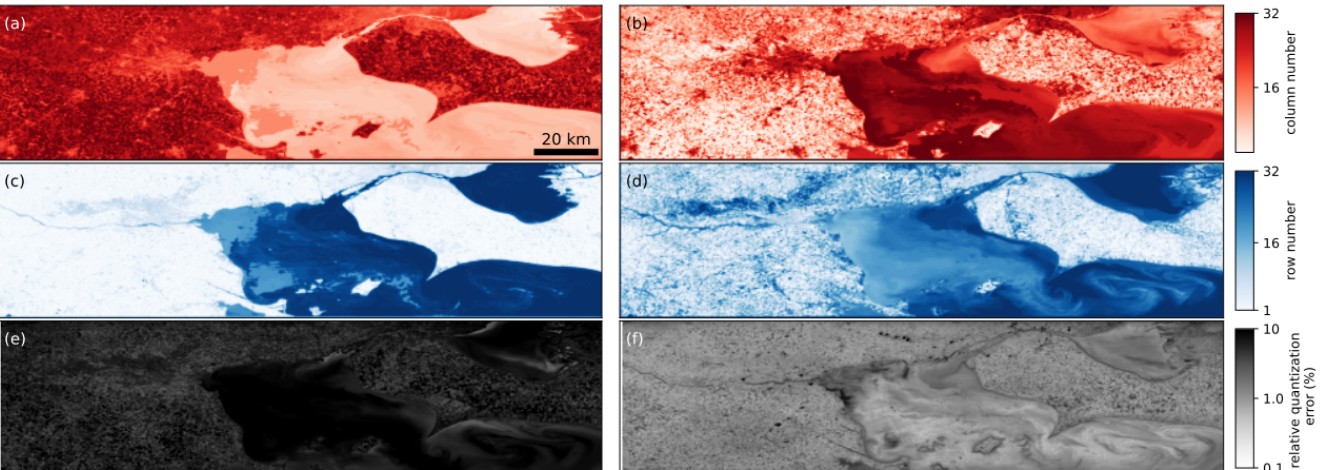

**Figure 10.** The HICO Lake Erie scene clustered according to the $32 \times 32 \times 5$ SOM calculated from the first North Sea scene (**left**) or the SOM calculated on itself (**right**), according to the (**a**,**b**) column number and (**c**,**d**) row number. (**e**,**f**) the relative quantization error.

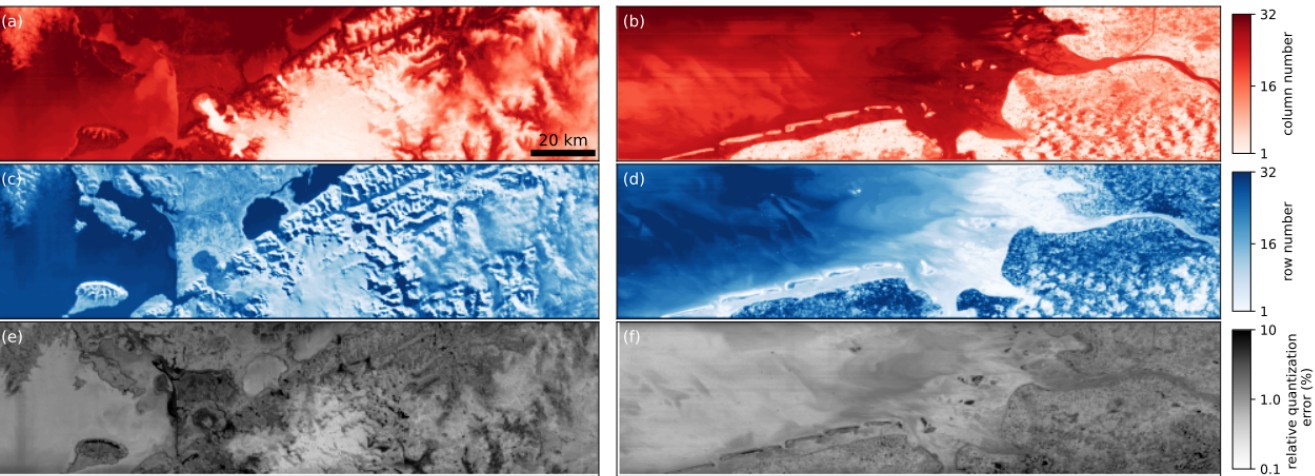

**Figure 11.** The Laguna San Rafael scene (**left**) and the second North Sea scene (**right**) clustered according to the SOM calculated on each, with the (**a**,**b**) column number and (**c**,**d**) row number. (**e**,**f**) the relative quantization error.

One additional concern regarding hyperspectral imaging on cube-satellites is that their small-form factor limits the size of the imaging optics, which, in turn, limits the signal-to-noise ratio of the collected data.

The effect of noise on SOMs is investigated by adding noise to the first North Sea image. The signal-to-noise ratio of the median observation (over all bands and pixels) is varied from 1 to about $10^4$ by adding noise to the scene, poissonian rather than gaussian in order to better approximate photon counting statistics. A $32 \times 32 \times 5$ SOM is initialized and trained on $10^5$ pixels from each noisy dataset, and is then evaluated on both the noisy and original datasets (Figures 12 and 13).

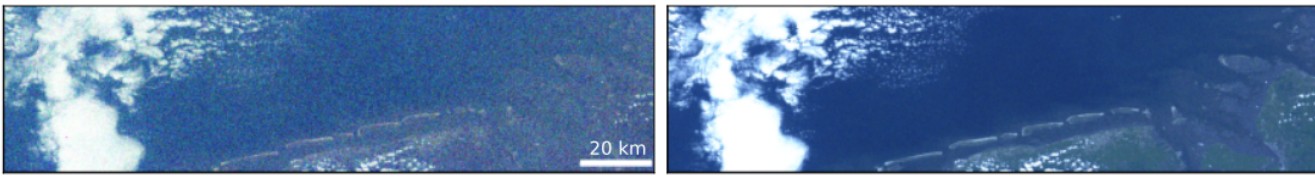

**Figure 12.** The first HICO North Sea scene adjusted to a signal-to-noise ratio of 1 for the median observation value (**left**). The same noisy scene after being reconstructed from data clustered by an SOM (**right**).

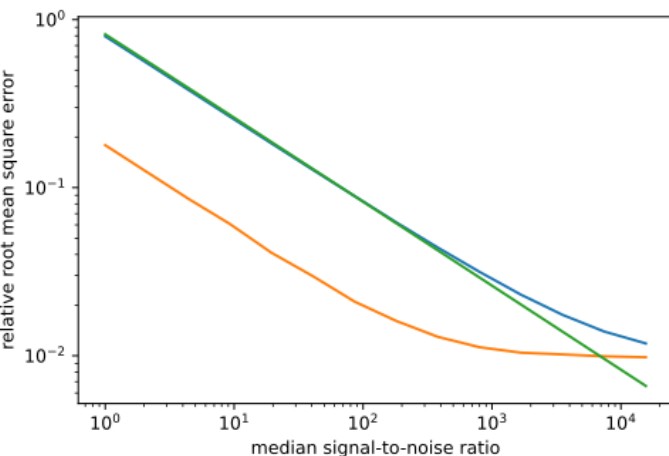

**Figure 13.** At small signal-to-noise ratios, the SOM-processed data (**yellow**) are a better estimate of the scene than the original noisy observations (**green**). While the SOM provides a good estimate of the original scene, the reconstruction provides a relatively poor estimate of the noisy scene on which it is actually trained (**blue**) because it does not capture the noise well.

It is found that the SOM-processed data better approximate the original, noiseless scene by a factor of almost 5 compared than the raw noisy observations, although this effect begins to vanish at signal-to-noise ratios above 1000. This noise reduction seems to originate from the way that each SOM node gives the average of several pixels, together with the selection of only five PCA dimensions [58,62]. As the averaging is non-local, it does not harm the spatial resolution of the scene. Thus the use of SOMs is not inhibited by noise and, in fact, may even alleviate some of the effects of noise.

### 3.3. Classification

Operationally, it is advantageous for hyperspectral images to not only be clustered, but also classified with labels that indicate distinct materials. For example, if the HYPSO-1 satellite could partition images into water, land, and clouds, then it could select only water pixels to downlink, which could ease the data budget. Similarly, Unmanned Aerial Vehicles (UAVs) will collaborate with HYPSO-1 by performing hyperspectral imaging under the clouds, which the satellite cannot observe. If HYPSO-1 downlinks a map of where the clouds as it images, the map could guide the UAVs to the locations they need to scan. Therefore, the capacity of SOMs to classify data is tested by labelling the nodes of a SOM and applying it to the test scenes.

Labels are applied to a few regions of the first North Sea scene, which is, in turn, used to label the nodes of an SOM that is used to apply labels to the whole scene. The broad classes of water, land, and cloud are used because distinguishing between those three components in within the expected skill level of satellite operators, who may not be experts in remote sensing. The regions of the scene which are selected as representative of the class are colored in Figure 14a. The pixels of each labelled class are mapped to the SOM, the counts of which are depicted in Figure 15a. These counts are then converted into probabilities using the neighborhood function of the SOM (Figure 15b). Finally, the probabilities are used to label the different nodes of the SOM (Figure 15c). The pixels in the scene are then classified according to their BMU (Figure 14c). Repeating the procedure with DR produces similar results (Figure 14e).

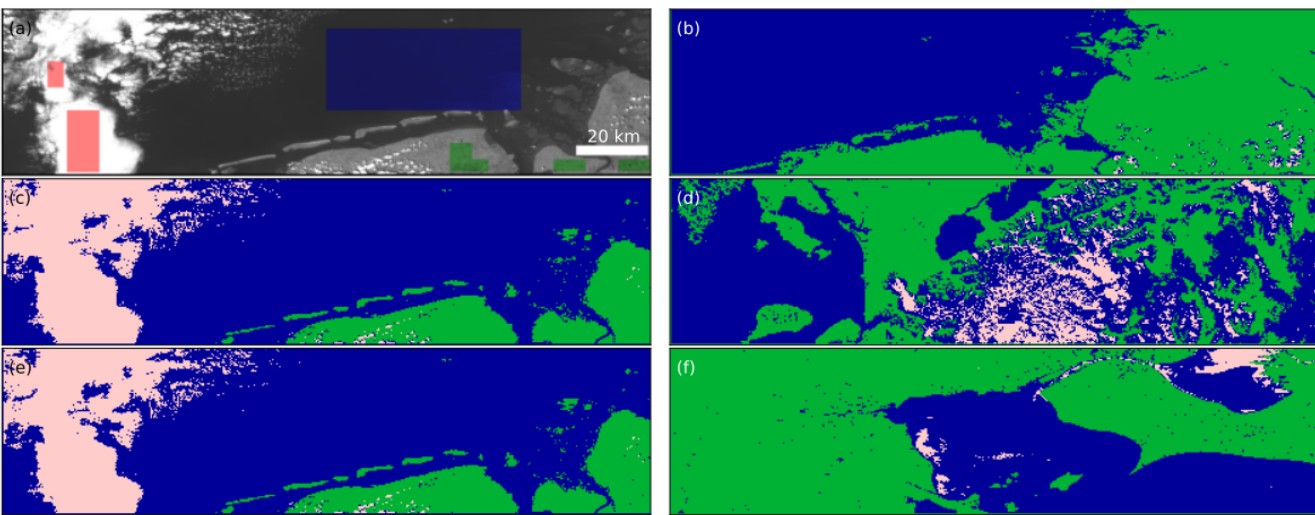

**Figure 14.** (**a**) The pixels assigned to the water (**blue**), land (**green**), and cloud (**red**) for the initial determination of labels. The labels assigned according to (**c**) the full 86-band SOM or (**e**) the five-band DR SOM. The DR SOM classification applied to (**b**) second North Sea scene, (**d**) the Chilean scene, and (**f**) Lake Erie.

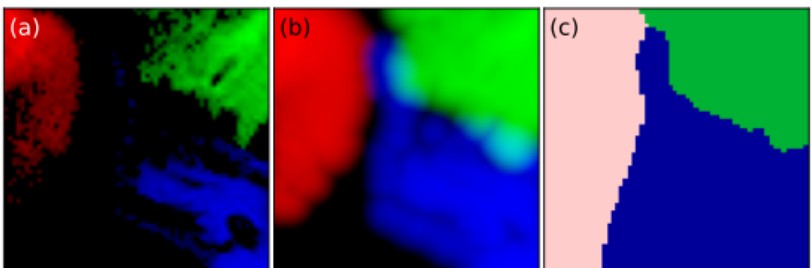

**Figure 15.** (**a**) The number of pixels assigned to each of the 64 nodes, scaled to the maximum of each class, for water (**blue**), land (**green**), and clouds (**red**), plotted logarithmically. (**b**) The estimated probability that a pixel in a given class is a member of an SOM node, plotted logarithmically with black = $10^{-15}$. (**c**) The class labels assigned to each node.

Classification fits more naturally with the first operational procedure (Table 3), because, in that plan, spatial information can be used to assist labelling different regions. Therefore, the $32 \times 32 \times 5$ SOM trained on the first North Sea scene is labelled according to the above procedure. Then, its node labels are used to classify the three other scenes (Figure 14, right column). The water is identified fairly well in all scenes, although, in the second North Sea scene, numerous water pixels are classified as land along the coast. In the Luguna San Rafael scene, the snowy mountainous region is classified as a mixture of water and clouds, perhaps because there is no snow in the training data. Moreover, in the same scene, clouds at the leftmost edge of the image are mistakenly classified as land. In the Lake Erie scene, a number of water pixels are classified as clouds. Thus, it is possible to transfer labelled SOMs to classify new scenes, but more pixels will be mislabelled than in the original data.

By testing the SOM classification on several common benchmark scenes, the preservation of the information in a scene is evaluated. The SOMs are benchmarked against support vector machines (SVMs), which are a state-of-the-art single-pixel classification technique. While more sophisticated classification techniques could be applied to data clustered by a SOM, for example, by including information about neighboring pixels, the focus here is on single-pixel techniques. SOMs and SVMs are compared on four different scenes: Samson, Jasper, Pavia University, and Indian Pines, with sizes $64 \times 64 \times 5$ on the first three scenes, and $128 \times 128 \times 5$ on the latter.

The training procedure is the same as that which is applied to HICO, shown above. Each scene is partitioned into a training set consisting of 90% of the pixels and a test set

consisting of 10% of the pixels. The effect of the size of the training set is the subject of an additional test. The SOM is trained on all of the pixels in each image, except for the test set. The labels are then applied using the training data. The same training/test partition is used to train the SVMs.

The SOMs classify the two simpler scenes, with 3 or 4 classes, about as well as the SVMs, but their performance relative to the SVMs degrades for the more complex scenes, with 9 or 16 classes (Table 7). The relative quantization error of the SOM clustering also increases with the complexity of the scene, with the SOM describing less than half of the variance on the Indian Pines scene.

**Table 7.** Performance of several classifiers on different scenes.

|  | **Samson** | **Jasper** | **Pavia** | **Pines** |
| --- | --- | --- | --- | --- |
| Mean Relative Quantization Error | 0.0289 | 0.2909 | 0.3693 | 0.5319 |
| SOM-Overall Accuracy | 0.9922 | 0.9690 | 0.8214 | 0.6635 |
| SVM-Overall Accuracy | 0.9845 | 0.9900 | 0.9178 | 0.8499 |
| SVM-Support vectors | 379 | 370 | 9354 | 4280 |

The effect of the number of labelled pixels on classification is tested on the Jasper scene, because it is more complex than Samson, but the number of different members are more balanced than the Indian Pines or Pavia University scenes, so that even a sample size of 16 pixels can contain all the different classes.

As expected, the overall accuracy of the SOM classification increases with the number of labelled samples (Figure 16). Whether the SOM is trained with only the labelled samples or both labelled and unlabelled samples matters little (solid vs. dotted red line). Both show an overall accuracy above 90% when a few hundred samples are used. For sample sizes below 100, the SOM and SVM show a similar overall accuracy, while the SVM shows more improvement with additional samples. Each procedure is repeated four times and averaged. This is evidence that operational procedure 2, in which the SOM is trained on a sampling of the raw data, will retain enough information for further image processing and scientific analysis (Table 4).

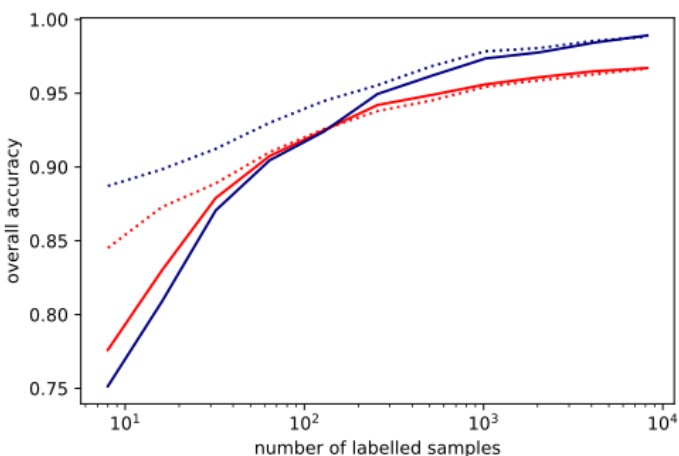

**Figure 16.** The overall accuracy for SVM (**blue**) and SOM (**red**) classification, with the classifier trained on different numbers of samples from the image Jasper. The dotted blue line shows OA for SVM trained with all 86 bands. The dotted red line shows SOM initialized by unlabelled samples.

On the final two scenes, the performance of the SOM degrades relative to the SVM classification, although it is also notable that the SVM uses more support vectors than on the prior scenes, but the SOM does not increase in complexity. The Indian Pines scene shows the worst performance in both the SOM clustering and classification (Figure 17). First, the mean relative quantization error is over 0.5, which indicates that the scene itself

is too complex to describe with an SOM of this size, even accounting for the fact that the SOM is slightly larger than those used before ($128 \times 128 \times 5$). In light of the very high relative quantization error, it is notable that the SOM is able to still achieve 66% overall accuracy in the classification.

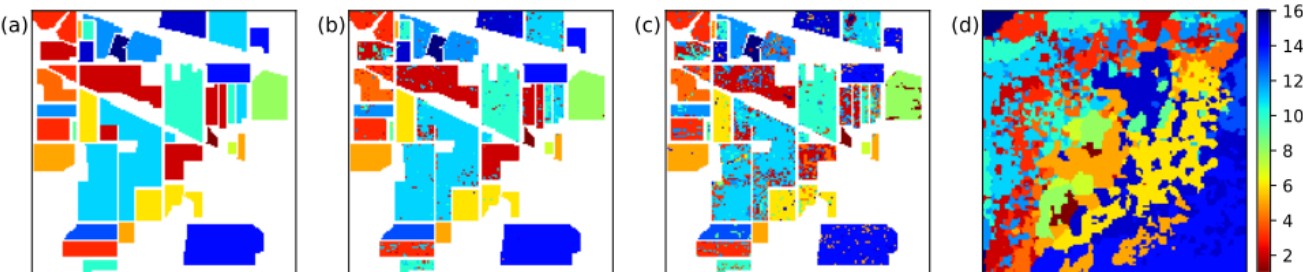

**Figure 17.** (**a**) The Indian Pines scene ground truth labels. (**b**) the scene classified by the SVM. (**c**) The scene classified by the $128 \times 128 \times 5$ SOM. (**d**) the class labels on the nodes of the SOM.

## 4. Discussion

The experiments with SOMs show that they are an appropriate tool for clustering hyperspectral data on-board a small satellite. The clustering can be used as interpretable compression, which reduces the size of a dataset by a factor over 100 relative to the raw data. By decreasing the downlink time, the SOMs can facilitate near-real-time hyperspectral imaging from small satellites. SOMs similarly increase the total throughput of the data that can be captured.

Of the two proposed operational procedures, the experiments show that the two-stage procedure, outlined in Table 4, will typically achieve a much lower relative quantization error than the single-stage procedure (Table 3). It seems reasonable to train an SOM with less than 1% total pixels. Still, the second procedure entails a more sophisticated operational plan, and requires either an operator to perform PCA and train the SOM between satellite passes, or an automatic procedure to train and assess the SOM before uplinking. Furthermore, the benefits of having an SOM trained on a random sample of points from the target image suggests that it may simplify the operations to bring the whole clustering procedure (including training) on-board the satellite. This would either require more computational power, perhaps by utilizing the programmable logic that is already on the satellite or by including a more powerful processing unit on a subsequent satellite in the HYPSO constellation. It is worth noting that SOMs have been developed for programmable logic previously [63–65].

Clustering by SOMs retains enough data for subsequent data-processing, as shown by the experiments with classification. However, it may be worth investigating ways to enable SOMs to retain specific information, as the SVMs outperformed the SOMs in terms of classification on more complicated scenes. Different DR techniques could lead to different information being retained in the SOM nodes.

## 5. Conclusions

Although simple, SOMs have numerous benefits when used as a clustering technique where computational power is limited. They are quick to both train and evaluate on different images, they have a small memory footprint, and their output is interpretable as nodes, which, for the hyperspectral data discussed here, can be seen as spectra. Moreover, the clustered scene can be simply reconstructed and, because it does not rely on spatial information, its output can provide spatially unbiased information to subsequent algorithms, and they can be trained from a few samples. Both operational procedures have been shown to be suitable for execution on a cubesat, although they bring different benefits. For these reasons, SOMs are potentially well-suited to the needs of hyperspectral imaging cubesats.

**Author Contributions:** Conceptualization, J.L.G.; methodology, A.S.D. and J.L.G.; software, A.S.D.; validation, A.S.D. and J.L.G.; writing—original draft preparation, A.S.D.; writing—review and editing, all authors; visualization, A.S.D. and J.L.G.; supervision, T.A.J. and J.L.G.; funding acquisition, T.A.J. All authors have read and agreed to the published version of the manuscript.

**Funding:** The research leading to these results has received funding from the NO Grants 2014–2021, under Project ELO-Hyp, contract no. 24/2020, and the Norwegian Research Council through the Centre of Autonomous Marine Operations and Systems (NTNU AMOS) (grant no. 223254) and the MASSIVE project (grant no. 270959).

**Data Availability Statement:** The data presented in this study are openly available from NASA Earthdata (HICO) or from www.ehu.eus/ccwintco/index.php/Hyperspectral_Remote_Sensing_Scenes, accessed on 15 July 2021, (Indian Pines and Pavia University) or from http://lesun.weebly.com/hyperspectral-data-set.html, accessed on 15 July 2021, (Jasper Ridge) or from https://opticks.org/display/opticks/Sample+Data, accessed on 15 July 2021, (Samson).

**Acknowledgments:** We would like to thank Sivert Bakken and Dennis Langer for looking over an early draft.

**Conflicts of Interest:** The authors declare no conflict of interest.

## Abbreviations

The following abbreviations are used in this manuscript:

| | |
|---|---|
| BMU | Best-Matching Unit |
| DR | Dimensionality Reduction |
| HSI | HyperSpectral Imager |
| HYPSO | HYPerspectral Smallsat for ocean Observation |
| PCA | Principal Component Analysis |
| SOM | Self-Organizing Map |
| SVM | Support Vector Machine |
| TOA | Top of Atmosphere |

## Appendix A. Computational Complexity of Clustering

Both the space and time complexity vary between the types of clustering algorithms discussed above. The size of hyperspectral datasets and the limited RAM in the onboard computer led the space constraints to be more limiting. In this appendix, a sketch of the space complexities of the algorithms are discussed and compared because they are most relevant to the comparison.

At each iteration of the $k$-means algorithm, the cluster index of each pixel must be stored before the new means are calculated. The size of the working memory is then $O(cn)$, where $c$ is the number of clusters. Even though self-organizing maps separate the training of the network from its execution, the space complexity is similar in both. At each iteration of training, the SOM algorithm requires an array of the distances between the input pixel and the nodes of the network, the neighborhood function evaluated at each node, and the SOM network itself. The first two arrays are of size $O(z^2)$, as they are simply a scalar at each node. The SOM network itself, on the other hand, is of size $O(z^2d)$. During execution, the second $O(z^2)$ array is no longer needed.

The memory requirements for the other clustering algorithms scale less well with the number of pixels under consideration. The first of them, spectral clustering requires storing the $n \times n$ affinity matrix. Several algorithms tailored to the large hyperspectral datasets have been developed [36,37]. These variants mitigate the large memory requirement by approximating the affinity matrix by a $n \times m$ matrix, where $m$, the number of anchor points, is free to be chosen. The authors of [37] chose to use $m \approx 0.1n$. The space complexity of model-based clustering algorithms which rely on spatial information (e.g., [41,42]) tends to be dominated by the array which contains information about the relationship between each pixel in its neighbors which is $O(pn)$, where $p$ is the number of neighbors considered. When spatial information is ignored, as can be done for gaussian mixture models

(e.g., [40]), the situation is similar to SOMs, as identified by [20]. Indeed, the space complexity is a product of the number of clusters and the mean and covariance matrix of the gaussian distribution which characterizes each, so that the space complexity is O($z^2 d^3$). Because the SOM had the lowest space complexity, being constant with respect to the number of pixels and having only linear dependence on the number of dimensions, it is chosen for a more in-depth investigation regarding its feasibility on-board a small satellite.

The time complexities of *k*-means, gaussian mixture models, and training the SOM all depend on the number of iterations, *t* which can be adjusted. For *k*-means it is O($cdnt$) because the distance from each pixel to each (original) mean is calculated as well as the new means. For the gaussian mixture model discussed in [42], it is O($cn^2 t$) because of the energy minimization sub-algorithm which is used. For the SOM, we showed above the training can be performed on the ground, so the execution time complexity is more important. Because it depends on the determination of the BMU which involves evaluation the distance between each pixel and all the nodes, the time complexity of evaluation it O($nz^2 d$). Note when algorithms require spatial information, as [42] does, the operational procedures for moving training to the ground no longer work. The most computationally expensive step of the spectral clustering methods is the computation of the affinity matrix, which considers the similarity between the anchors and every pixel in the dataset, giving a complexity of O($dnm$). The SOM is the fastest among these, and could possibly be made even faster by using the spatial ordering of the SOM to reduce the time complexity to $O(dnz)$ [66].

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
