# Peer review of "Self-Organizing Maps for Clustering Hyperspectral Images On-Board a CubeSat"

_remotesensing, doi:10.3390/rs13204174_

Round 1
Reviewer 1 Report
In this paper, the authors present a strategy for using self-organizing maps (SOM) to classify hyper-spectral images on a satellite. The SOM machine learning technique is chosen because it requires relatively low memory and processor and separates training and execution so that the algorithm can be split between the ground segment and the spacecraft. they tested SOM on HYPSO-1 hardware and used these performance tests to derive size requirements based on runtime.
The SOM technique can be performed on the satellite within computational time constraints and the resulting clustering retains enough information for further processing of the image, such as classification.
I have the following remarks:
1- the paper is poorly organized: the summary is not clear, you must clearly show the problematic studied, your own contributions and the results of simulations and comparison. Also for the introduction which contains three sections, you must organize it by separating the state of art which must contain the problematic and the proposed contributions and related works.
2- The authors are invited to mathematically study the complexity of the SOM technique and to compare it with the other classification methods. the same goes for the computation time.
3- I have a problem regarding the use of PCA for dimensionality reduction if the first major axes do not represent enough information.
4-For a better reading, I invite the authors to introduce who describes the SOM technique in the case of supervised and unsupervised learning.
Author Response
We thank you for your feedback and have copied the response to your comments here as text. We recommend looking at the pdf, which also shows the responses of the other reviewers, as well as the changes highlighted in the text.
"In this paper, the authors present a strategy for using self-organizing maps
(SOM) to classify hyper-spectral images on a satellite. The SOM machine learning technique is chosen because it requires relatively low memory and processor and separates training and execution so that the algorithm can be split between the ground segment and the spacecraft. they tested SOM on HYPSO-1 hardware and used these performance tests to derive size requirements based on runtime. The SOM technique can be performed on the satellite within computational time constraints and the resulting clustering retains enough information for further processing of the image, such as classification.
I have the following remarks:
1- the paper is poorly organized: the summary is not clear, you must clearly
show the problematic studied, your own contributions and the results of simulations and comparison. Also for the introduction which contains three sections, you must organize it by separating the state of art which must contain the problematic and the proposed contributions and related works."
We have significantly revised the introduction, at the reviewer’s request,
by separating sections on related works and proposed contributions, by adding
a paragraph about on-board processing to the related works, and by adding
additional details about the experiments to the proposed contributions.
In addition, we revised the abstract to focus on our new results regarding
how SOMs can be incorporated operationally.
"2- The authors are invited to mathematically study the complexity of the
SOM technique and to compare it with the other classification methods. the
same goes for the computation time."
We appreciate the reviewer’s interest in the complexity of the SOM technique. We have added an appendix with some discussion of the space and time
complexities of the different clustering techniques.
"3- I have a problem regarding the use of PCA for dimensionality reduction
if the first major axes do not represent enough information."
Figure 8 shows the percentage of the variance which is captured by a given
number of PCA components in the test image. The first component consists of
just under 90 percent of the variance, and the first 5 together account for over
99 percent of the variance.
We have also added the text 'its only hyperparameter is number of dimensions,' to clarify why we chose PCA as a dimensionality reduction technique.
"4-For a better reading, I invite the authors to introduce who describes the
SOM technique in the case of supervised and unsupervised learning."
In the introduction, we have emphasized the reference Riese et al 2020 which
describes supervised, semi-supervised and unsupervised Self Organizing Maps
for hyperspectral data.

Reviewer 2 Report
The use of SOMs on multispectral and hyperspectral satellite imagery isn't new, so novelty factor of this work is low. The authors present the application of SOMs for nanosatellite image preprocessing which is restricted by hardware and bandwidth specifications. The use of custom C language code to speed up execution seems adequate for the purpose. Precision of the results are clearly explained and well presented. However no new algorithms or enhancements that could represent a better way of speeding up or reducing memory footprints on such algorithms are proposed, and just standard metrics for SOMs are employed.
Author Response
We thank you for your feedback and have copied the response to your comments here as text, but we recommend looking at the pdf, which also shows the responses of the other reviewers, as well as the changes highlighted in the text.
"The use of SOMs on multispectral and hyperspectral satellite imagery isn’t
new, so novelty factor of this work is low. The authors present the application
of SOMs for nanosatellite image preprocessing which is restricted by hardware
and bandwidth specifications. The use of custom C language code to speed up
execution seems adequate for the purpose. Precision of the results are clearly
explained and well presented. However no new algorithms or enhancements that could represent a better way of speeding up or reducing memory footprints on such algorithms are proposed, and just standard metrics for SOMs are employed."
We appreciate that you find the results “clearly explained and well presented”
The novelty in the research comes from the adaptation of the SOM for
edge computing and the connection the analysis has with different operational
procedures. There have not been any studies investigating how to operationalize SOMs on an hyperspectral imaging satellite, or any other similar platforms.
Some of the abstract and introduction have been rewritten to emphasize that the focus of this article is not on new developments to the SOM architecture, but rather a comparison of two different ways that SOMs can be incorporated in to a satellite mission and an evaluation of the effectiveness of the clustering, both with respect to compression and classification.
We admit that SOMs have been studied before, but find that the comprehensive literature on them is a benefit for incorporating them into an operational procedure, rather than a detriment.
We would also like to note that even the basic application of SOMs to hyperspectral data is still an active research topic, as indicated by Riese et al 2020 and Wong et al 2019.

Reviewer 3 Report
The manuscript was explored the capability of Self-Organizing Maps on labelled and unlabelled hyperspectral data. I found that the paper is fine and reliable. I only concern about the length of the manuscript.
From line 59 to 150 – Did authors relay need all this information to introduce the problem statement. I think there are normal information in sections 1.1 and 1.2 and should be reduced.
You don’t need to provide a definition for every single concept, especially for the known concepts. For example (Line 207), it is quite known what is dimensionality reduction and the used method to reduce the reduction is PCA in remote sensing community and you don’t need to restate the definition again. I suggest to start the paragraph from line 210 “Principal component analysis (PCA) is used …”.
Line 220 – again, classification is defined elsewhere above in the introduction and you repeat the same definition here.
Line 229 – like the classification section. Here you repeated the same discussion.
The results section is far too long, and the number of figures is a too large (25 figures?). Results section and the number of figures must be reduced. You don’t need such description for every section and figure.
Figures and tables should be embedded after they are referred in the text not before.
Maps do not contain scales. This is basic in cartography.
Some words were repeated twice. Please revise through the entire manuscript and delete the repeated words.
Author Response
Thank you for your helpful comments. Please see the attachment, which contains both our response to your comments and to the other reviewers, as well as a copy of the revised document with changes highlighted in blue.

Round 2
Reviewer 1 Report
The authors have made all the changes requested. For this purpose, I suggest the acceptance of the paper
Reviewer 2 Report
The authors have been more specific about the limitations of their research in the revised version of the paper.